# Decolonizing Brazil through Science Fiction: *Bacurau* and Brazilian Empowerment

Lidia Zuin 

Visual Arts, State University of Campinas, São Paulo 13083-970, Brazil; lidiazuin@gmail.com

**Abstract:** The objective of this article is to present an overview of the changes in Brazilian science fiction throughout the past decades, which culminated in the international and national success of the film *Bacurau* and the emergence of the new subgenres amazofuturismo and sertãopunk in continuation of Causo's tupinipunk. These subgenres are bringing to the plate topics such as nationalism, decolonization, racism, social justice, and globalization, while using pop formulas and, thus, engaging an audience that is finding in those titles an opportunity to address the current political scenario. This paper suggests that *Bacurau*, amazofuturismo, and sertãopunk are new manifestations that are paving the way for contemporary Brazilian artists, who are seeking to overcome what Rodrigues' called "the mongrel complex", by updating certain proposals already made by tupinipunk in the 1990s. As the country celebrates the 100th anniversary of the Modern Art Week, this article proposes that contemporary Brazilian science fiction is echoing contemporary humanities agenda topics, such as decoloniality and politics of recognition, in response to the past four years of Jair Bolsonaro's government.

**Keywords:** Brazilian science fiction; Bacurau; tupinipunk; decoloniality; science fiction

## 1. Bacurau as Catharsis

Released in 2019, the film *Bacurau* (dir. Kleber Mendonça Filho and Juliano Dornelles) has achieved a worldwide box office of over USD 3 million, while using a Hollywoodian formula (often compared to Quentin Tarantino's style), but adapted to the Brazilian context. That same year was also the moment when the Oscar-awarded movies *Parasite* (2019, Dir. Bong Joon-ho) and *Joker* (2019, Dir. Todd Phillips) were released and critically acclaimed for their social-political commentary. In Brazil, these three movies became a triad that summarized contemporary feelings about culture, politics, and economy; however, since *Bacurau* was a national title that added the "Brazilian seasoning" to the Blockbuster formula, it became an emblematic or even cathartic film for Brazilians who had experienced the recent election of far right wing president Jair Bolsonaro.

Some of the reasons behind this acclamation lie in the fact that *Bacurau* tells an empowering story about a community in the northeast of the country, a region often referenced for its background of scarcity and its history of social banditry, represented by the Cangaço[1]. Mendonça and Dornelles, thus, merge past and contemporary references, while blending fictional genres such as the western, thriller, and science fiction, in order to tell the story of a village overwhelmed by American hitmen, who were hired by a local politician after realizing he would not be supported by the villagers in the upcoming elections.

Technological artifacts are featured in *Bacurau* as ambiguous metaphors about globalization and colonization, though they are also appropriated by the community and "hacked" according to their need. Right in the first minutes of the movie, we watch a boombox car display in its LED screen a series of CCTV recorded images of a local criminal named Pacote (played by Thomás Aquino), who decided to quit banditry. Smartphones are used by villagers at school for classes and for communication, whereas the same device is employed by the hitmen to stalk the village and control drones in a gamified carnage

scheme. In fact, the hitmen assume that the Bacurau dwellers do not know or understand technology, and this is why they disguise a drone as a flying saucer, supposing that the villagers would take it for an UFO. While this works as a not too subtle commentary about the controversies of capitalism, globalization, post-colonialism, and racism, the villagers do not mistake the drone for an UFO, but indeed recognize the device in disguise.

I suggest that the ambiguity of technology in *Bacurau* echoes Donna Haraway's (1985) concept of "informatics of domination", since the villagers use these same devices as a means to subvert the hierarchies of domination and, thus, adopt a role that is represented by the author's meaning of cyborg: a being of agency that can use technology to politically confront the status quo. However, when it comes to the assumption that the villagers were ignorant or even primitive compared to the foreigners, this reflects not only on xenophobic and evolutionist assumptions, but it mostly hints at the inferiority complex observed by the writer Nelson Rodrigues in the 1950s, when he proposed the concept of "Complexo de Vira-Lata" (Mongrel Complex). In Rodrigues' words: "For 'mongrel complex', I understand the inferiority in which Brazilians put themselves, voluntarily, in contrast to the whole world. The Brazilian is an upside-down Narcissus who spits on his own image. Here is the truth: we do not find personal or historical pretexts for self-esteem" (Rodrigues 1993).

Oliveira (2019) describes the mongrel complex as the idiosyncrasy that marks the way Brazilians voluntarily feel inferior to each and every one, but, when gifted with a victory, for instance in a World Cup, we finally see ourselves in "the right to walk down the street as if we were the rulers of the world". The author argues that, in order to make the colonizers the sole narrators of the story, there was a historic suppression of the many voices that would better represent the Brazilian identity and culture. This in turn has produced the assumption that Brazilians are inferior, since their story is told (or rather suppressed) from the conquerors' point of view.

Although Rodrigues originally addressed the term in the context of football, it turns out that the "mongrel complex" is a transversal issue in the country and has also found manifestation in science fiction. While Emilia Freitas' *A Rainha do Ignoto* (Freitas 1899) was later coined the first Brazilian science fiction book, the genre had its first traction in the literary market when mainstream authors such as Chico Buarque de Holanda used tropes of speculative fiction as a means to address topics that were forbidden by the military dictatorship in the 1960s. Later on, in 1982, a second wave of authors began to draft possible paths for a Brazilian science fiction (SF).

Back then, most SF references used by Brazilian writers were anglophone titles, a fact that resonated in the debate around pursuing a more universal writing style or the development of a more nationalist, unique approach to Brazilian SF. This became clearer in the works of Jorge Luiz Calife, who was in contact with Arthur C. Clarke and is known to have inspired the writing of the sequence to *2001: A Space Odyssey*. While his style was closer to Anglophone SF, it was Ivan Carlos Regina's controversial *Anthropophagic Manifesto of Brazilian Science Fiction* (1988) that promoted a more original approach to the genre, by using Oswald de Andrade's *Anthropophagic Manifesto* (1928)[2] as a reference.

It was in this context that the author Roberto de Sousa Causo created the "tupinipunk" subgenre as an anthropophagic, Brazilian version of the American cyberpunk (Sterling [1985] 1986), which was coined after merging the words "tupiniquim" and "cyberpunk". Back in 1989, when Causo proposed this concept, "tupiniquim", which is the name of a Brazilian indigenous tribe, was popularly used as a synonym to "Brazilian" and also a metaphor for Brazilian copycats that were as good as the original, or at least good enough, though not ideal (Causo 2013).

Such play of words reflected the mindset of a SF subgenre that held an

> ... iconoclastic attitude, sensuality, mysticism, politicization, and a Third World perspective. Tupinipunk works are less worried with true exploration of hard science fiction than their American counterpart; and the junkie culture is less determinant. This science fiction subgenre would never be called 'a radical hard science fiction', one of the first labels of cyberpunk, but its approach towards

technology and society is closer to satire and postmodern playfulness than to consistent [technological and scientific] extrapolation. ([Causo 2013](), pp. 226–27)

While *Bacurau* fulfills some of the characteristics that denote tupinipunk, the film also updates the subgenre by highlighting topics of the contemporary agenda in Brazilian politics. The election of president Jair Bolsonaro in 2018 has ignited a countercultural response in the country, since the politician is known for his hate speeches against minorities, the deliberate neglect of environmental issues, and the blatant corruption superposed by conservative ideals that have been feeding a persistent legion of followers and fake news[3].

In this article, I aim to discuss how *Bacurau* inaugurates a new era for Brazilian SF as authors incorporate afrofuturism[4] into their agenda, as well as new subgenres such as amazofuturismo and sertãopunk. These movements were created in response to this new countercultural moment and in the pursuit of a better and more diverse representation of the complexity and multiplicity of Brazilian national identity through speculative fiction, by breaking with common stereotypes and eliminating the historic point of view enclosed in the colonizer's perspective. It is through publications such as the *Irradiative Manifesto* (2015), the Sertãopunk manifesto and anthology, as well as the founding of new publishing houses that we observe that Brazilian SF has been following a path that is less universalist, but still in dialogue with globalization and its effects. This is what I will present in the following pages.

## 2. Tupinipunk, Amazofuturismo, and Sertãopunk: Anthropophagy and Emancipation

First proposed as visual concepts in the art of the illustrators Vitor Wiedergrün[5] and João Queiroz, the Brazilian SF subgenres sertãopunk and amazofuturismo have been explored by writers who are working on decolonizing narratives which seek for a balance between the effects of globalization and the pursuit of what it means to be Brazilian in the 21st century. Nevertheless, it is important to stress that these efforts are being made with a conscious goal of not falling prey to an exacerbated nationalism and a stereotypical representation of local identities. One example of this is found in the release of the e-book *Sertãopunk* by authors G.G. Diniz, Alan de Sá, and Alec Silva in ([de Sá et al. 2020]()), which included a manifesto and two short stories that inaugurated the subgenre. In a similar manner, Mario Bentes' short story *Pajemancer* also inaugurated the aesthetics and tropes of amazofuturismo in literature, as the narrative is featured in the anthology *2084: Mundos Cyberpunks* (2018).

Examples such as these run in parallel to current theories in social sciences, such as is the case of the concept of politics of recognition proposed by Charles [Taylor]() ([1994]()). The author lists three kinds of recognition in politics: (1) politics of universalism, which aims for equal recognition of all persons in their common humanity; (2) politics of difference, which is only one dimension of a politics of recognition, and it emphasizes the uniqueness of specific cultural features; (3) recognition of concrete individuality, which is used in the context of loving care and affection. As these concepts already resonate in post-coloniality and decoloniality studies, they have also found their way into the agenda of the so-called fourth wave of Brazilian SF writers ([Oliveira 2021]()), who are working on the update (or ultimately the overcoming) of Rodrigues' "mongrel complex" and emancipation from the colonial stereotypes still present in Brazilian culture and arts.

Moreover, colonialism has also taught Brazilians to follow a lifestyle and culture that have nothing to do with the particularities of the country. In movies such as *Carlota Joaquina, Princesa do Brazil* (1995, Dir. Carla Camurati), there is a comedic approach to the way that the Portuguese court is characterized, while living under the scorching sun of Rio de Janeiro. Although wigs, heavy makeup, and layered outfits had nothing to do with the Brazilian climate and cultural particularities, locals were educated to follow this etiquette and experience a feeling of inferiority for not being able to adapt to such habits.

Likewise, the attempt to "whitewash" the country through the raising of children with fairer skin was a means of eliminating what was then considered inferior—that is, the Indigenous and African ascendance in the population. [Oliveira]() ([2019]()) mentions that

social Darwinism played an important role, since slavery was abolished in the country in 1888, since the doctrine suggested a correlation between a supposed superiority of a civilization and its race, an idea that consequently supported the pursuit of the eradication of the black, indigenous, and multiracial peoples (*mestiços*), even on a genetic and not solely phenotypic sense. According to Oliveira Jr., this was the "unnamed" background that supported Rodrigues' proposition of a mongrel complex[6].

In order to overcome such a rooted prejudice, Oliveira Jr. suggests that affirmative action should be developed and proposed by the Public Power, as well as the media, so that these institutions may pursue a multiculturalist approach to the issue. However, the author stresses that multiculturalism in this context does not have the same meaning as once proposed by Zygmunt Bauman, who understood it as a rather conservative project, which aims at "the transformation of social inequalities, a phenomena whose approval is generally highly unlikely, under the disguise of 'cultural diversity', that is, a phenomena that deserves universal respect and careful cultivation" (Oliveira 2019, p. 12).

In face of such a mistaken interpretation, Oliveira Jr. proposes that people follow a "crude version" of multiculturalism, one which does not mask social inequalities, but "recognizes the peaceful coexistence of several cultures in the same ambient, mixes them and reaches [a state of] becoming". By citing Deleuze and Parnet's understanding of "becoming", Oliveira Jr. suggests that "becoming" here proposes originality, a creation that does not come from the "assimilation" of foreign references, but rather from its "deglutition", as suggested by Oswald de Andrade in his *Anthropophagic Manifesto* (1928).

That is what happens in *Bacurau*: the film portrays the resistance of the villagers in face of foreign intervention, turning the threat into an opportunity to "devour" these external elements, instead of simply and unquestionably assimilating them. The movie, therefore, attempts to break with the common sense belief that colonization happened without resistance; much as Andrade took the episode of the devouring of Bishop Sardinha[7] as an inspiration for his *Anthropophagic Manifesto* and a representation of how indigenous people reacted to colonization.

While *Bacurau* does not feature scenes of cannibalism, similar violence is portrayed in a nod to gore movies[8], though these elements are not used purely to cause shock, but they work as a symbol to debunk the commonsense idea that indigenous people did not react to colonization, as well as the subsequent ways in which capitalism and globalization are currently pursuing a new kind of colonization. By combining both references from gore movies and the effects of late capitalism, the Mexican philosopher Sayak Valencia analyzed the phenomenon through the concept of "gore capitalism"[9].

In a similar fashion, Dias (2020) claims that *Bacurau* uses speculative fiction to catalyze the desire for vengeance and resistance that was shared by Brazilians at the time of the movie's release in 2019, one year after the election of Bolsonaro. However, *Bacurau* is also concerned to show the other face of colonialism in Brazilian culture, by featuring a couple of Brazilian bikers that were hired by the American hitmen to investigate the village.

When questioned by the foreigners about their decision to work for them, and therefore betraying their own people, the couple explains that they were born in the southern part of Brazil, a region known to be wealthier and "more developed", besides being a region where most European descendants live[10]. It is, however, when the couple affirms that they are white that the hitmen respond that, no matter how fair is their skin, they will always be Latinos, and thus inferior.

What follows next is that, after a child is murdered by the hitmen, the villagers ask for help to Lunga (played by Silvero Pereira), a queer *cangaceiro* who initiates a guerilla plan, in which the villagers use the historic weapons displayed in a small local museum (which resembles the Museum of Canudos[11]) to punish the invaders. By the end of the movie, the hitmen are decapitated in the same way the *cangaceiros* historically decapitated their enemies (Rubens 2010). Still, in spite of giving a nod to a historical fact, the aesthetic utilized in *Bacurau* puts this scene closer to Tarantino's proposal in *Inglorious Basterds,* rather

than anything canonically explored by previous Brazilian movie directors of the Cinema Novo movement; thus, assuming a more pop approach in its visuals and narrative.

With all this considered, *Bacurau* can, therefore, be understood as a contemporary attempt to incorporate the ideas posited by Ivan Carlos Regina in his *Anthropophagic Manifesto of Brazilian Science Fiction* (Regina 1988), a work that was written in a time when Brazilian artists were trying to find their identity in science fiction. Here, I translate the full text:

> Man went to the stars to find himself and found only the void, void, void.
>
> He found out that in the interior of all suns, night is hidden, and with it, its ancestral enemy, the darkness.
>
> His travel companions are death, pain, laugh, sex, misery, happiness, love, boredom, solitude, hopelessness, tiredness and laziness.
>
> There is a pyramid of useless objects crossing existence: a microwave, a plastic bottle, a kilo of ether, a nylon shirt, a razor blade. Everyday objects.
>
> We do not propose the dialectic of society, but the aesthetic of novelty.
>
> Man hates God and loves the robot. His destiny is to destroy perfection and create aberration.
>
> The totem was the first machine of man.
>
> We want to be an explosion of shape and a revolution of content. A supernova in the sky of conventionality.
>
> Happiness is casting out nines.
>
> Technology is, ultimately, man's neurotic attempt to replace all his human components for artificial ones, thus creating a world where he is the least possible to be responsible.
>
> A boitatá[12] with eyes of Cesium is lurking the Brazilian Highlands[13],
>
> By dealing only with machines, science fiction has turned into a genre of scenarios, a mockery of *vaudeville*, sterile and inconsequent.
>
> We did not come to criticize the function of machines, but to propose the aesthetic of man.
>
> We need to urgently swallow, after Bishop Sardinha, the laser gun, the mad scientist, the good little alien, the invincible hero, the space folding, the bad little alien, the lady with perfect legs and a nut for a brain, the flying saucer[14], which are so distant from Brazilian reality as the furthest of the stars.
>
> Brazilian science fiction does not exist.
>
> The copy of the foreigner model creates children with wide eyes, old book freaks, writers with no readers, neurotic men, escapist literatures, absurd books that are summed up by their covers and mental poverty, intellectual colonies which seek, in a grotesque imitation, to recreate the *modus vivendi* of technologically developed countries.
>
> The national science fiction cannot come in tow with the rest of the world. Either we achieve quality or we disappear.
>
> The Brazilian literary production, in the SF genre, in spite of its reduced list of works, has such a horrifying mediocrity.
>
> A headless mule spewing radioactive fire through its breath[15].
>
> We emulate technologies without knowing them.
>
> A Saci Pererê[16] ponders with a vanadium prosthetic, chews cassava, grinds paçoca[17] and burps enriched uranium.
>
> Happiness is casting out nines.

Man proves, every day, that he does not deserve technology.

We want to wake up the iconoclast that lies in every Brazilian chest.

Death to the machine worshippers.

A green and yellow hick devours hamburgers, destroys satellites, swallows weapons and shreds technologies.

An indigenous person will come down from a bright colorful star.

Although published decades ago, Regina's manifesto is echoing once again in contemporary Brazilian SF, as new authors are precisely discussing how the genre can subvert international and colonizing stereotypes by retrieving topics and information that were being eclipsed through new subgenres such as amazofuturismo and sertãopunk. In the past some authors have used canonical literature such as Machado de Assis' *Dom Casmurro* to publish science fiction spin-offs, where UFOs ([Manfredi 2010](#)) interact with the original lore and characters. More recently though, instead of relying on the canon, and precisely wanting to create something anew, authors are finding ways to fulfill Regina's call to action by bringing up less obvious folklore and native cultures, as well as debunking stereotypes in a way that I suggest has been incentivized by the success of *Bacurau* and its critical approach to speculative fiction.

While Regina's manifesto plays with stereotypical folkloric characters and produce (Saci Pererê, paçoca, cassava etc.) and globalized references (mad scientist, satellites, hamburgers etc.), what subgenres such as amazofuturismo and sertãopunk want to do is precisely reveal the "other side of the coin", by proving that Brazilian culture goes much beyond what is already crystallized in stereotypes. With sertãopunk[18], the creators of the subgenre wanted to show that Brazilian northeastern culture and history are much more than dry landscapes, social banditry, and poverty.

However, even before that, Roberto de Sousa Causo's proposal for tupinipunk had already pursued many of the ideas presented by Regina in the manifesto. Though the subgenre was created in 1989, Causo consolidated its definition in his thesis *Ondas nas Praias de um Mundo Sombrio: New Wave e Cyberpunk no Brasil* (2014), where he dedicates a full chapter to describing the concept that has already been adopted by academia, for instance through the work of M. Elizabeth [Ginway](#) ([2005](#)) (author of the thesis *Brazilian Science Fiction: National Myths and Nationhood in the Land of the Future*), Juan Ignacio Muñoz Zapata from the University of Montreal, who wrote the thesis *Le cyberpunk vernaculaire de l'Amérique latine: dystopies, virtualités et resistances*, and Karina Vázques in her essay *Brazilian Cyberpunk and the Latin American Neobaroque: Political Critique in a Globalized World*. According to Causo, the novel *Silicone XXI* ([Sirkis 1985](#)), by Alfredo Sirkis, is the first tupinipunk book, as it includes characteristic elements such as analogical and non-digital media, cultural and religious syncretism[19], international conspiracies, and sexuality.

[Causo](#) ([2013](#), p. 265) also argues that while cyberpunk could be considered a "radical post-modernist fiction that collects references from science fiction and hard-boiled fiction, the junkie culture, and punk rock, tupinipunk tries to look avant-garde by retrieving the attitudes of the most known local avant-garde movements", which include the anthropophagic Brazilian modernist movement of the 1920s and 1930s. Despite the fact that the Brazilian modernists took inspiration from European Dadaists, Surrealists, and Futurists, their work was adapted to the local context, as well as it deals, with an aesthetic that radically reacted to the "artistic conventionalism and the colonial residues in the Brazilian society, which they nicknamed as 'passadismo'[20]".

What is more, [Causo](#) ([2013](#)) states that, unlike cyberpunk, tupinipunk barely refers to hard science and the main topics and leitmotifs of science fiction in its works. These tropes are here approached in the form of satire or citation; thus, turning the references into a pastiche. Tupinipunk is, therefore, a subgenre that uses religion or mysticism as its signature of syncretism, and these topics are approached as a means to initiate social, cultural, and political debates. Nevertheless, according to Causo, in terms of writing style, tupinipunk narratives use a post-modernist aesthetic, in which the plot, the characters,

and topics approached are secondary or even inexistent. The subgenre prefers to focus on external contexts that are "underlined by identity and relationship ambiguities in which the very nature of the 'self' loses its usual parameters and becomes a slippery pursuit that rarely results in the transformation of the characters" (Causo 2013, p. 266).

Causo mentions Robert Scholes' concept of "fabulation"[21] to infer that tupinipunk points "to the arbitrary nature of narratives, and it doesn't aim to be a true representation of reality". Moreover, he says that tupinipunk dialogues with an audience and literary tradition that only superficially enters into contact with science fiction and its mannerisms; henceforth, making the subgenre closer to the Brazilian modernist movement and its syncretism than to the traditions of science fiction. Consequently, Causo argues that tupinipunk's appropriation of syncretism rather highlights the "typical adaptation of the Brazilian political tradition and its social discourse of conciliation—in which basic tensions of a contradictory country are only adapted, and not seriously or fairly approached" (Causo 2013, p. 252).

Syncretism was also a denomination used to describe the Brazilian pre-modernist literary movement (from 1910 to 1920), which was already discussing topics such as race and culture, as seen in the work of Silvio Romero (1851–1914). Causo stresses that, between 1882 and 1903, Romero was already using the words "mestiço"[22] and "mestiçagem" to state that the "mestiço" was "the original Brazilian" or "Brazilian by excellence". His contributions have resonated in Andrade's *Anthropophagic Manifesto* and, consequently, in Regina's SF manifesto and tupinipunk. Causo claims that, since Romero published his essays, the ideological meaning of "mestiço" has actually migrated to the plane of syncretism, which is more often used to describe religion and folklore; thus, denoting its cultural and artistic effort to represent the idea of "mestiçagem"—a transformation which Causo says that, in its turn, attenuated Romero's racialist commentaries that were tinged with social Darwinist ideas[23].

However, Causo also cites Homi K. Bhabha's arguments about "cultural diversity" and "cultural difference" as a means to defend tupinipunk's tendency to reinforce racial stereotypes. He affirms that tupinipunk often features " the stereotyping of the sexuality of afrodescendants, referring to the size of their genitalia, large nostrils, and the salutation of a sexual superiority between black people". The same goes for sexual freedom, which he attributes to a libertarian sense inscribed in Donna Haraway's (1985) definition of cyborg: "The cyborg is a creature in a post-gender world; it has no truck with bisexuality, pre-oedipal symbiosis, unalienated labour, or other seductions to organic wholeness through a final appropriation of all the powers of the parts into a higher unity".

To Causo, the reproduction of such stereotypes does not impede tupinipunk in criticizing colonialism. He cites Ginway's interpretation of tupinipunk, where she says that "the deliberate politics, primitivism and eroticism are basic principles of tupinipunk, and that distinguishes it from the North American counterpart. In its representation of race, sexuality, urban space and multimedia, tupinipunk uses the body as space of cultural resistance". She follows:

> It is precisely the exotic image [of Brazil], combined to the type of technology that the Brazilian cyberpunk takes to the extreme, which allows it to challenge the literary hegemony of the American cyberpunk and make a self-conscious parody of the elitist notions of the Brazilian high culture. In this sense, tupinipunk is related both to the Brazilian modernist tradition of a cultural cannibalism as well as to a [post-colonial] sensibility. (apud Causo 2013, p. 272)

In the case of *Bacurau*, it is not completely incorrect to say that the film uses some stereotypical representation of northeastern Brazilians, but that could possibly be a resource to fit the movie into a pastiche of the Hollywoodian formula. On the other hand, the narrative itself subverts the clichés by adding new layers to the characters. In the case of Lunga and Pacote, they are criminals that are not portrayed after a macho trope associated with the northeastern culture (as seen in the expression "cabra macho"[24]) or with the

stereotypical representation of criminals in reference to *cangaço*. In fact, it is implied during the movie that Lunga and Pacote possibly had an amorous relationship before, whereas Lunga himself offers a queer interpretation of a social bandit ("cangaceiro") in his characterization[25].

Controversially, Causo (2013) also claims that classic tupinipunk narratives written by second wave authors use stereotypes for satirical reasons, and they do not necessarily invite the reader to reflect, or present an arc of moral transformation for the characters. Although stereotypes could be used in tupinipunk to raise political debate, Causo seems to infer that this is not mandatory, especially in the case of second wave authors. In this sense, even if we consider that *Bacurau* fits most of the characteristics which describe tupinipunk, the movie still proposes an update to the subgenre, by revamping it with topics and propositions that are aligned to the political and activist agenda held by contemporary Brazilian SF authors. Once again, the future is not what it used to be: stereotypes and prejudice are no longer material for jokes and entertainment; it is not safe anymore to expect that audiences will take these tropes into question when the country elects Bolsonaro as president.

Since the early 2010s, Brazilian SF authors have been working on ways to invite fellow readers and writers to reconsider their choices and tropes. Through the *Irradiative Manifesto*[26] (Anotsu and Vieira 2015), authors Jim Anotsu and Vic Vieira appealed for more diversity in Brazilian speculative fiction, after a historical analysis on how the genre has been dominated by white cis, heterosexual male authors. Consequently, the rise of sertãopunk and amazofuturismo, as well as the popularization of afrofuturism, are showing that authors are responding with more localized, diversified, and complex representations of Brazil's multiple identities. *Bacurau*, by its turn, has also functioned as a "escape valve", as suggested by Dias Jr., since the film incites a cathartic reaction in the audiences that opposed the election of Bolsonaro.

During the past decades, the foundation of smaller publishing houses, such as Lendari, Wish, Fora do Ar, Clepsidra, Draco, Panda Books, and others, has increased the opportunity for publication of new and already experienced writers, including cartoonists, who are now finding more opportunities to publish their comics in partnership with writers. The use of crowdfunding platforms such as Catarse has also ignited a market populated by digital influencers (booktubers), who promote literature in social media platforms such as Instagram, Twitter, YouTube, TikTok, and Twitch. Likewise, collectives such as Boreal are also bringing to the plate the discussion and inclusion of LGBTQI+ authors of speculative fiction.

Unlike the beginning of the genre's formation in the 1980s, the discussion now is not about following foreign trends and emulating anglophone literature in Brazilian Portuguese, but rather how contemporary Brazilian authors can create and ignite their own movements, while still taking part of global trends and movements. One successful example is the anthology *Solarpunk–Histórias ecológicas e fantásticas em um mundo sustentável* (2013), which was translated to English and published by World Weaver Press (a publishing house specialized in Solarpunk and Global South speculative fiction) in 2018. In a similar fashion, increasingly, more Brazilian authors have had their works featured in international magazines.

What is more, the creation of CoFutures at the University of Oslo paves the way for an international academic environment that is dedicated to contemporary global futurisms and speculative fiction. As described in their website, CoFutures is "involved in research in different sectors including theoretical research, technology research, policy research, artistic research, as well as production or support for transmedia artistic work including fiction, films, and video games". Headed by Professor Bodhisattva Chattopadhyay, the research center features the PhD candidate Patrick Brock who is working on a research work about Brazilian afrofuturism "and its implications for cultural production and regeneration, racial identity, and resistance to authoritarianism" through the analysis of books, films, videogames, and artworks.

In other words, movements such as tupinipunk, amazofuturismo, and sertãopunk, as well as Brazilian afrofuturism, are igniting this change for a post-colonialist approach to science fiction. This has been seen in the work of authors Fábio Kabral (2017), Ale Santos, and Lu Ain Zaila, for instance. In the case of Kabral, his novel *O Caçador Cibernético da Rua 13* features elements of African Brazilian religions, while Lu Ain Zaila presents her take on Afrofuturism in the short stories of the collection *Sankofia: Breves histórias sobre afrofuturismo*. Ale Santos (2019) has also established his work by being chosen as one of the finalists for the Jabuti Awards in 2020 for his book *Rastros de Resistência,* which addresses the history of past African kingdoms that were erased or suppressed by colonialism and post-colonialism. More recently, Santos (2021) announced the publication of *O Último Ancestral*, an Afrofuturist science fiction novel, in which he presents a dystopian future in Brazil, while addressing topics such as favelas, religious diversity, Carnaval, and issues such as racial segregation and structural racism.

In 2021, the Museum of Tomorrow at Rio de Janeiro promoted, in partnership with Santander, a science fiction workshop mentored by authors including Ale Santos, Alexey Dodsworth, Lu Ain-Zaila, Lidia Zuin, and Julie Dorrico, whose work is focused on literature produced by Brazilian indigenous peoples. It is worth mentioning that afrofuturism and amazofuturismo have also found expressive manifestation in the music industry, with artists such as Xênia França, Jonathan Ferr, KatúMirim, Kunumí MC, and more. In face of such events, I suggest that it is safe to assume that *Bacurau* has opened up a new era for Brazilian science fiction and fantasy, which is following its success and formula for a new perspective on speculative fiction.

## 3. Conclusions

In Brazil, the year of 2022 brings new elections, which hold the possibility that both Bolsonaro and Lula will be running for president. At the same time, the country celebrates the centenary of the Brazilian Modern Art Week, which is ultimately expressed and represented by Andrade's *Anthropophagic Manifesto* and reflected in Regina's *Anthropophagic Manifesto of Brazilian Science Fiction.* The fact that Brazilian science fiction is experiencing a new "wave" in this same period is no coincidence, but I suggest that it is rather a consequence of and response to the past four years of Bolsonaro's government—an assumption that is also made by Dias Jr. in his review of *Bacurau.*

What this paper concludes is that Brazil is opening up a new era for speculative fiction and a different approach to the implications of (post) colonialism in the 21st century. By resonating with the ideas that inaugurated the first decades of the 20th century modernist movement, contemporary Brazilian artists are showing through their work a new take on identity, culture, globalization, and an increasingly feasible and tangible way to overcome Rodrigues' mongrel complex. Unlike Causo's tupinipunk, this new wave of writers are concerned with breaking with stereotypes and showing more of the diversity of the many "Brazils" who are unknown even to natives; thus, showing other ways in which Brazilians can find resonance with local culture, rather than feel ashamed or diminished in the face of foreign stimuli; whether caused by pop culture or by remnants of colonialism.

While *Bacurau* portrays a near-future version of a Brazilian northeastern village, it does not ignore the advances of globalization and the thorough adoption of emerging technologies in the country. Without falling prey to dystopian stereotypical representations, the movie presents a means through which the margins can revamp technology coming from the center; thus, fulfilling the hypothesis (or call to action) made by Donna Haraway in the 1980s. Likewise, contemporary Brazilian artists are paving the way for the overcoming of the "mongrel complex", not by intensifying a sense of patriotism and nationalism[27], but by dialoguing with globalization.

By acknowledging the past and swallowing the future, *Bacurau*'s narrative is a metaphorical invitation to post-colonial action; it is about acknowledging history (even if it takes the population to preserve it in a museum, like in the film), but it is no excuse to be stuck in time or to comply with evolutionist ideals that will keep people stuck in the

mongrel complex. *Bacurau* shows that the future is not just about building skyscrapers in the hinterlands, but using smartphones to transcend the physical barriers and bring back to the community new, cannibalized versions of the world. With these thoughts, I conclude this paper with the provocative question asked by the French artist Makan Fofana (2021) in *La Banlieue du Turfu*: why do we need to go to the center to live the future and not simply bring it to the margins? As presented here, *Bacurau* and Brazilian contemporary SF artists seem to be answering this question with interesting suggestions.

**Funding:** This research received no external funding.

**Institutional Review Board Statement:** Not applicable.

**Informed Consent Statement:** Not applicable.

**Data Availability Statement:** Not applicable.

**Conflicts of Interest:** The author declares no conflict of interest.

## Notes

[1] Cangaço was a phenomena of social banditry that happened in the wilderness of the northeast of Brazil between the 18th and mid-20th centuries. Members of this movement (known as cangaceiros) wandered in groups, crossing states, and attacking cities, where they looted, murdered, and raped people. In spite of their means, the Cangaço movement is often and popularly seen as an act of self-defense of the country people, who suffered from grave social inequality and the inability of the State to rule. Virgulino Ferreira da Silva, popularly known as Lampião, was one of the main leaders of the movement, which was named after the word "canga", the name of a wooden piece used to lock the cattle with a cart or plow.

[2] Andrade's *Anthropophagic Manifesto* was written in inspiration and reflection of the Brazilian Modern Art Week in 1922.

[3] For a more comprehensive profile of Bolsonaro, I suggest reading The Guardian's article "Who is Jair Bolsonaro? Brazil's far-right president in his own words" (2018). https://www.theguardian.com/world/2018/sep/06/jair-bolsonaro-brazil-tropical-trump-who-hankers-for-days-of-dictatorship (accessed on 26 April 2022).

[4] Coined in 1993 by Mark Dery, the term "afrofuturism" expanded during the late 1990s as Alondra Nelson started to shape a series of conversations about the concept. In her words: "Afrofuturism has emerged as a term of convenience to describe analysis, criticism, and cultural production that addresses the intersections between race and technology. Neither a mantra nor a movement, Afrofuturism is a critical perspective that opens up inquiry into the many overlaps between technoculture and black diasporic histories".

[5] Originally, Wiedergrün proposed the term "cyberagreste", which was later criticized and revised by the creators of sertãopunk.

[6] While the use of "mongrel" in Portuguese refers to dogs with mixed breeds, multiracial people denotes individuals of more than one race or ethnicity. For a long time, these two concepts were used to compare human races and dog breeds, which is a racist premise, as argued by Norton et al. (2019).

[7] Dom Pedro Fernandes Sardinha, or Pero Sardinha, (1496–1556) was a Portuguese priest and first bishop of Brazil. On 16 July 1556, he and his crew were shipwrecked and captured by the Caeté people, and even though he had indicated he was a great prelate of the Portuguese and a priest, he was still slaughtered with a mace and devoured along with his companions.

[8] The participation of the actor Udo Kier in *Bacurau* is also considered a nod from the directors to his history in gore and horror films.

[9] In her book Capitalismo Gore, Valencia makes a comparison between the way capitalism shatters and devalues human lives and how narco groups work, claiming that the latter are only more intense and explicit in their actions, but both function after similar premises.

[10] The state of São Paulo, located in the southeastern part of Brazil, is the richest state in the country. It concentrates 10% of the whole national GDP. In the beginning of the 20th century, various European and Asian families migrated to Brazil. According to the national census of 1920, there were over 1 million immigrants living in Brazil, 90% concentrated in the southern part of the country. More at: https://www.scielo.br/img/revistas/rsp/v8s0/03t4.gif (accessed on 26 April 2022)

[11] A museum created by the merchant Manoel Alves in 1971 in the city of Canudos, Bahia. As a popular museum, the institution did not receive any institutional support, but Alves was so impressed to learn about the history of the Canudos War that he decided to keep and expose every object that was related to that episode. In the collection, there are old sewing machines, chests, cartridges and bullets, shotguns and revolvers, and machetes and sheaths that were supposedly used during the war. With that being said, it is important to stress that both the real Museum of Canudos and the fictional museum of Bacurau were not created and maintained by public institutions, but rather kept by civilians that wished to safeguard the memory of the event and the history of the region.

[12]    Boitatá is a Tupi-Guarani term used to designate the willow fire, which by extension has inspired the creation of some of the first mythological creatures registered in the history of Brazil.

[13]    The manifesto was written one year after the Goiânia accident, which occurred on 13 September 1987, in the city of Goiânia. The accident involved radioactive contamination coming from a radiotherapy source found at an abandoned hospital site. Several people handled the material, resulting in four deaths and 249 people being diagnosed with contamination by radioactivity. Brazilian Highlands is the region where the city of Goiânia is situated.

[14]    *Bacurau* makes a commentary about this trope during the disguised drone scene, as I discussed before.

[15]    Regina refers here to the Headless Mule, a mythical character in Brazilian folfklore. In most tales, it is told the story of the ghost of a woman that has been cursed by God for her sins and, thus, turned into a fire-spewing headless mule. Here Regina adds the detail of a radioactive fire, probably inspired by the Goiânia accident.

[16]    Saci Pererê is a character of Brazilian folklore known as a one-legged black or bi-racial young prankster, who smokes a pipe and wears a magical red cap, which enables him to disappear and reappear wherever he wishes (usually in the middle of a dust devil).

[17]    Brazilian candy made of peanut.

[18]    Sertãopunk was proposed after the article "Amazofuturismo e Cyberagreste: por uma nova ficção científica brasileira" (Zuin 2019) caused controversy. Authors Alan de Sá, Alec Silva and G.G. Diniz have shared their views on the article on Medium posts and YouTube videos, which preceded the release of the manifesto and short story anthology "Sertãopunk: Histórias de um Nordeste do Amanhã" (de Sá et al. 2020).

[19]    Causo (2013) employs the term syncretism here in the sense of a "combination of ideas and habits with different origins, sometimes antagonic, which produce confusing visions of a complex totality. The [Brazilian] condition of a third-world country is transparent in the Brazilian syncretism, [it appears] as a cultural appropriation that does not offer a deeper understanding of the contributing cultures".

[20]    A literal translation of "passadismo" could be "past-ism", that is, an appeal and attachment to the past. The Brazilian Modern Art Week caused controversy at the time because some art critics such as the writer Monteiro Lobato (1917) reacted negatively to the aesthetic proposed by modernist artists such as Anita Malfatti, for example. According to him, her art (and therefore Brazilian modernist art) did not follow the canon and would be a much better fit for the "walls of psychiatric hospitals" rather than in museums or galleries.

[21]    In *Fabulation and Metafiction* (1979), Scholes used the term fabulation to indicate novels which violate standard novelistic expectations by adding experimental elements. These could be in the way the subject matter is approached, the writing style and format, as well as the blending of everyday life and fantastic, mythical, or even nightmarish components. As an example, Scholes uses fabulation to address Kurt Vonnegut's science fiction.

[22]    A person of mixed racial ancestry. In the Brazilian context, this could mean mixed European, African, and Native American ancestry.

[23]    "Mestiçagem cultural" (Xavier 2008) is a term that has been used by academia to describe Brazilian "anthropophagic" style of appropriating cultural references, blending genres, and creating something new and particular. While in this context, the words "mestiço" or "mestiçagem" are not used with a negative connotation, it is indeed possible to see how they could echo Rodrigues' "mongrel complex" as the term encompasses the idea of miscegenation and thus "impurity". However, contemporary anti-racism activists and authors have suggested the appropriation of words that used to be slurs, in order to empty their negative meaning and turn them into affirmative terms. That is the case for the word "preto", for instance (Vicenzo 2021). In this case, Causo suggests that using the word "mestiço", in affirmation to an identity, is a means to oppose to the social Darwinist ideas that tinged Romero's essays when the term was used.

[24]    More about masculinity in the Brazilian northeastern culture can be found in Trotta (2013).

[25]    It is worth mentioning that Silvero Ribeiro, the actor who portrays Lunga, is known in Brazil for his activism in the LGBTQ+ community, as well for his drag performances. Silvero was also born in the northeastern part of Brazil and one of his political causes is the discussion about the stereotypes connected to the region where he was born and raised.

[26]    Available online: https://manifestoirradiativo.wordpress.com/eng/ (accessed on 26 April 2022).

[27]    In fact, this is a trope that was already criticized and pointed out even in pre-modernist Brazilian literature, such as it is the case of the novel *Triste Fim de Policarpo Quaresma* (Barreto 1915), by Lima Barreto.

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
