# Peer review of "Decolonizing Brazil through Science Fiction: Bacurau and Brazilian Empowerment"

_humanities, doi:10.3390/h11030063_

Round 1

Reviewer 1 Report

  • A brief summary

The Author discusses the movie Bacuraru showing that it may be considered an attempt to 'decolonize' Brazil. The movie is analyzed in the context  Brazilian science fiction - its new subgenres and much older ideas. Overall, the Author claims that science fiction in Brazil is engaged with renegotiating national identity.

  • Broad comments

The  Author needs to specify her/his aims. We are given to understand the paper is going to revaluate Brazilian science fiction . I still wonder why the author does not analyze (but just rewrites)  Regina's Manifesto ll. 187-231.  There are some interesting points that are just glossed over here.

  • Specific comments

lines 35 Please explain your reference. Why Haraway's manifesto is important for you in this context? Why not Bruce Sterling's texts?

line 36-48 You should be more specific why this summary should make us realize that we are dealing with the "Mongrel Complex"? I am not sure what this  phrase means;

line 52-55 A confusing passage: the term "science fiction" was coined in the 1920s  and Freitas wrote her book in 1899 (of course it is possible and often advisable to retrospectively code earlier texts as SF but we need to know on what grounds); please reflect on how you use the terms: "science fiction", "speculative fiction", a "Second Wave";

line 56 what is meant by "Anglo-Saxon"?

line 73 please refer to Bruce Sterling's "The New Science Fiction", the first manifesto of  the cyberpunk movement;

line 168 "unforgettable" here suggests that the value judgement is based on  subjective feelings only;

lines 265 the critic's name is Robert Scholes; please refer to his definition of "fabulation" (how Scholes uses "fabulation" e.g. to discuss Kurt Vonnegut's science fiction)

lines 286 What do you mean by "essays that were tinged with Social Darwinist ideas"? This point should be elaborated on and/or linked to the "Mongrel Complex" part of your paper

line 342- 396 Parts of the CONCLUSION should be moved to the main body of the paper. New ideas are oddly located near the end of the article, where they come as a kind of afterthought.

Author Response

Thank you for your review and suggestions.

  • More commentary on Regina's manifesto was added.
  • I am using Haraway's reference instead of Sterling's texts because I am following her scholarly definition of cyborg as a posthuman existence in a world freed from what she calls "informatics of domination".
  • Definition of Mongrel complex was moved right after its first mention in the text, as suggested.
  • Adjusted the part which mentions Freitas' book as the first science fiction book in Brazil, as suggested.
  • Changed the term Anglo-Saxon to Anglophone.
  • Sterling's "The New Science Fiction" manifesto added as reference.
  • Changed the paragraph where "an unforgettable scene" was mentioned.
  • Added the information about fabulation according to Scholes
  • Reformulated the paragraph on social darwinism and the term "mongrel complex" for better comprehension
  • Moved parts of the conclusion to the main body as suggested
  • Other changes were made in response to other reviewers' commentary

Reviewer 2 Report

Although I enjoyed many aspects of this article, I’m afraid that I cannot recommend it for publication. The material is interesting and the author makes a number of valuable observations. However, it suffers from the following main weaknesses:

  • The argument is too unfocused. The essay touches on a number of themes - including the legacies of modernismo in Brazilian SF, post-colonialism, Afrofuturism – without exploring any of them with the depth necessary for an academic article.
  • The argument is under-referenced. While several theorists are discussed, these references feel a bit “dropped in” – without the necessary exposition or integration with the argument of the essay.
  • The essay is a bit too uncritical about the concept of national identity.
  • There is not enough analysis of the film itself – especially at the level of cinematic techniques and strategies.

Author Response

Thank you for your comments and suggestions.

  • Unfortunately, I am not qualified enough to make an analysis of the film from the perspective of cinematic techniques and strategies as I do not have any specialization in these areas. This is the reason why I am concentrating on the semiotic elements of the movie instead.
  • More description to terms were added as requested and well as better specification of the objectives of the paper, which is not to go into detail about the subgenres but present a "photograph" of the current situation of Brazilian science fiction.
  • Regarding mentioning authors instead of going deeper into their theories and concepts, since there are several references being used in this paper, I wanted to avoid the excess of footnotes or digressions for concepts that should be already familiar for the reader who will first be interested in checking the paper. Still, the revised version includes more contextualization as suggested.
  • Other changes can be seen in the text as part of other reviewers'  recommendations.

Reviewer 3 Report

This reviewer is familiar with Russian and Eastern European sf traditions, and the material in this article was almost entirely new to me. The value of this article is that it brings the originality and "alternative pathways" of Brazilian speculative fiction into focus. The article is well-documented and its claims convincing. The connection between current Brazilian pop-gore-sf-punk and aspects of early 20th-cen Brazilian modernism rings true. The political valence of the currently Brazilian scene is explored well in this article. 

Author Response

Thank you for your evaluation and commentary! I am submitting the paper once again as it was slightly modified after receiving the evaluation of other reviewers. Please see the attachment.
